# The association of chest radiographic findings and severity scoring with clinical outcomes in patients with COVID-19 presenting to the emergency department of a tertiary care hospital in Pakistan

Raima Kaleemi[1], Kiran Hilal[1]*, Ainan Arshad[2], Russell Seth Martins[3,4], Avinash Nankani[4,5], Haq TU[1], Sundas Basharat[1], Zeeshan Ansar[6]

1 Radiology Department, Aga Khan University Hospital, Karachi, Pakistan, 2 Department of Medicine, Aga Khan University Hospital, Karachi, Pakistan, 3 Medical College, Aga Khan University Hospital, Karachi, Pakistan, 4 Research and Development Wing, Society for Promoting Innovation in Education, Karachi, Pakistan, 5 Dow University of Health Sciences, Karachi, Pakistan, 6 Microbiology Department, Aga Khan University Hospital, Karachi, Pakistan

* kiran.hilal@aku.edu

## Abstract

### Introduction

While chest x-rays (CXRs) represent a cost-effective imaging modality for developing countries like Pakistan, their utility for the prognostication of COVID-19 has been minimally explored. Thus, we describe the frequency and distribution of CXR findings, and their association with clinical outcomes of patients with COVID-19.

### Methods

All adult ($\geq$ 18 years) patients presenting between 28th February-31st May to the emergency department of a tertiary care hospital in Pakistan, who were COVID-19 positive on RT-PCR with CXR done on presentation, were included. A CXR Severity Score (CXR-SS) of 0–8 was used to quantify the extent of pulmonary infection on CXR, with a score of 0 being negative and 1–8 being positive. The patients' initial CXR-SS and their highest CXR-SS over the hospital course were used for analysis, with cut-offs of 0–4 and 5–8 being used to assess association with clinical outcomes.

### Results

A total of 150 patients, with 76.7% males and mean age 56.1 years, were included in this study. Initial CXR was positive in 80% of patients, and 30.7% of patients had an initial CXR-SS between 5–8. The mortality rate was 16.7% and 30.6% patients underwent ICU admission with intubation (ICU-Int). On multivariable analysis, initial CXR-SS (1.355 [1.136–1.616]) and highest CXR-SS (1.390 [1.143–1.690]) were predictors of ICU-Int, and ICU-Int was independently associated with both initial CXR-SS 5–8 (2.532 [1.109–5.782]) and

**Data Availability Statement:** Data cannot be shared publicly because of confidentiality based restrictions placed by the ethics review committee of Aga Khan University Hospital. Data are available from the AKUH Institutional Data Access / Ethics Committee (contact via erc.pakistan@aku.edu) for researchers who meet the criteria for access to confidential data.

**Funding:** The author(s) received no specific funding for this work.

**Competing interests:** The authors have declared that no competing interests exist.

highest CXR-SS 5–8 (3.386 [1.405–8.159]). Lastly, age (1.060 [1.009–1.113]), initial CXR-SS (1.278 [1.010–1.617]) and ICU-Int (5.047 [1.731–14.710]), were found to be independent predictors of mortality in our patients.

## Conclusion

In a resource-constrained country like Pakistan, CXRs may have valuable prognostic utility in predicting ICU admission and mortality. Additional research with larger patient samples is needed to further explore the association of CXR findings with clinical outcomes.

## Introduction

With the outbreak of an unknown pneumonia in Wuhan, China, in December 2019, a new human coronavirus, SARS-CoV-2 (severe acute respiratory syndrome coronavirus-2), roused the attention of the entire world. COVID-19 (coronavirus disease 2019), the potentially fatal disease caused by SARS-CoV-2, was declared a pandemic by the World Health Organization in March 2020 [1].

As the pandemic unfolds, healthcare systems worldwide continually seek to determine the role of imaging in the diagnostics, management, and prognostication of COVID-19. The use of ultrasound (US) has been explored previously in the diagnostics of other infectious diseases [2]. Some studies have considered the emerging role of portable US imaging of the lungs in COVID-19 detection, describing ultrasound findings such as B-lines, consolidation and thickened pleural lines [3]. However, US has its limitations, as it is highly operator-dependent and difficult to perform in patients with severe disease in an intensive care unit (ICU) set-up. Moreover, the specificity of ultrasound in patients with COVID-19 presenting with early disease has also proven to be significantly lower as compared to chest x-ray and computerized tomography scans [4]. The utility of CT (computerized tomography) scans for COVID-19 imaging is being extensively explored [5]. However, the sheer influx of suspected and confirmed COVID-19 cases presenting to hospitals means that there is a major burden on radiology departments, posing immense challenges for infection control in the CT suite. The American College of Radiology notes that CT decontamination required after scanning patients with COVID-19 may disrupt radiological service availability, and suggests that portable chest x-ray (CXR) may be considered to minimize the risk of cross-infection [6].

Additionally, while CT scans have a good sensitivity in detecting subtle changes due to COVID-19, their cost and other practical considerations limit their utility in developing countries such as Pakistan [7–9]. In Pakistan, the number of confirmed COVID-19 cases has crossed 395,000 as of 29[th] November 2020, and the country is entering a second wave of COVID-19 infections [10]. At the Aga Khan University Hospital, a tertiary care hospital in Pakistan, patients presenting to the emergency department (ED) with COVID-19 symptoms are assessed with a baseline set of investigations that include RT-PCR (reverse transcriptase polymerase chain reaction) and CXR. Plain film radiography of the chest (CXR) is relatively inexpensive and is widely available as an imaging modality in the smaller healthcare centers of Pakistan. Though the radiographic features of CXRs in patients with COVID-19 have been described [11], there is a scarcity of literature discussing the association of CXR findings with clinical outcomes, particularly in our setting. Thus, in this article we aim to describe our experience of CXR imaging in patients with COVID-19, and explore the association between CXR findings and clinical outcomes.

## Materials and methods

### Setting and sample selection

This retrospective study was conducted between 28th February-31st May 2020 at the Aga Khan University Hospital, a tertiary care hospital in Pakistan. For this retrospective study, ethical approval was obtained from the institutional review board of the Aga Khan University Hospital (Reference Number: 2020-4774-10611). We were granted a waiver of informed consent as this was a retrospective study and all patients were discharged from the hospital. No personal identifiers were included in data collection, and records were anonymized to the statistician. Adult patients ($\geq$ 18 years) of either sex who presented to the emergency department with suspected COVID-19 based on clinical symptoms (fever, dry cough, dyspnea etc.), travel history, or positive RT-PCR (reverse transcriptase polymerase chain reaction), were considered for inclusion. Patients were enrolled if they were positive for COVID-19 based on RT-PCR and underwent CXRs. Any patient with a known history of any other pulmonary infection, or who was already admitted to the hospital due to some other disease, was excluded from the study.

### Data collection

Patients' data collected included clinical data accessed through the online patient care records and radiographic data as follows:

- **Clinical Data**: These included demographics and pre-existing comorbids (diabetes, hypertension, chronic kidney disease, malignancy, etc.) for all patients. For inpatients, additional variables included intensive care unit (ICU) admission with intubation (ICU-Int), hospital outcome (mortality vs. healthy on discharge/left against medical advice/discharged on request), and hospital length of stay (LOS).

- **Radiographic Data**: Two senior radiologists, with experience of more than 8 years in chest imaging, reviewed the initial and follow-up CXRs of all patients in the study. In cases of disagreement, further review by a third radiologist, who had more than 10 years of experience, was obtained. Radiographic features described through consensus of the radiologists included laterality, centrality, zonal location, and type of infiltrates. All radiologists were blinded as to the diagnosis and current clinical condition of the patient.

### Radiograph severity scoring

For quantitative measurement of extent of pulmonary involvement, a CXR Severity Score (CXR-SS) was calculated by adapting and simplifying the Radiographic Assessment of Lung Edema (RALE) score proposed by Warren et al. [12]. A score of 0–4 was assigned to each lung depending on the extent of involvement by consolidation or alveolar/interstitial infiltrates (0 = no involvement; 1 = <25%; 2 = 25–50%; 3 = 50–75%; 4 = >75% involvement). The scores for each lung will be summed to produce the final CXR-SS out of a maximum of 8 [11]. An example of the application of this scoring system is shown in **Fig 1**. A total score of 0 was considered as negative, while any score from 1–8 was considered as positive. This CXR Severity Score has been used before to quantify the extent of pulmonary infection in COVID-19 [11]. We adopted this scoring system to evaluate its prognostic value for clinical outcomes of COVID-19 in our setting. For the purpose of analysis, we used the patients' initial CXR-SS and highest CXR-SS. The initial CXR-SS was the score given to the patients' first CXR upon presentation to the hospital. The highest CXR-SS was the score with the greatest value amongst all the CXRs of a patient during their hospital course, which included both initial and subsequent

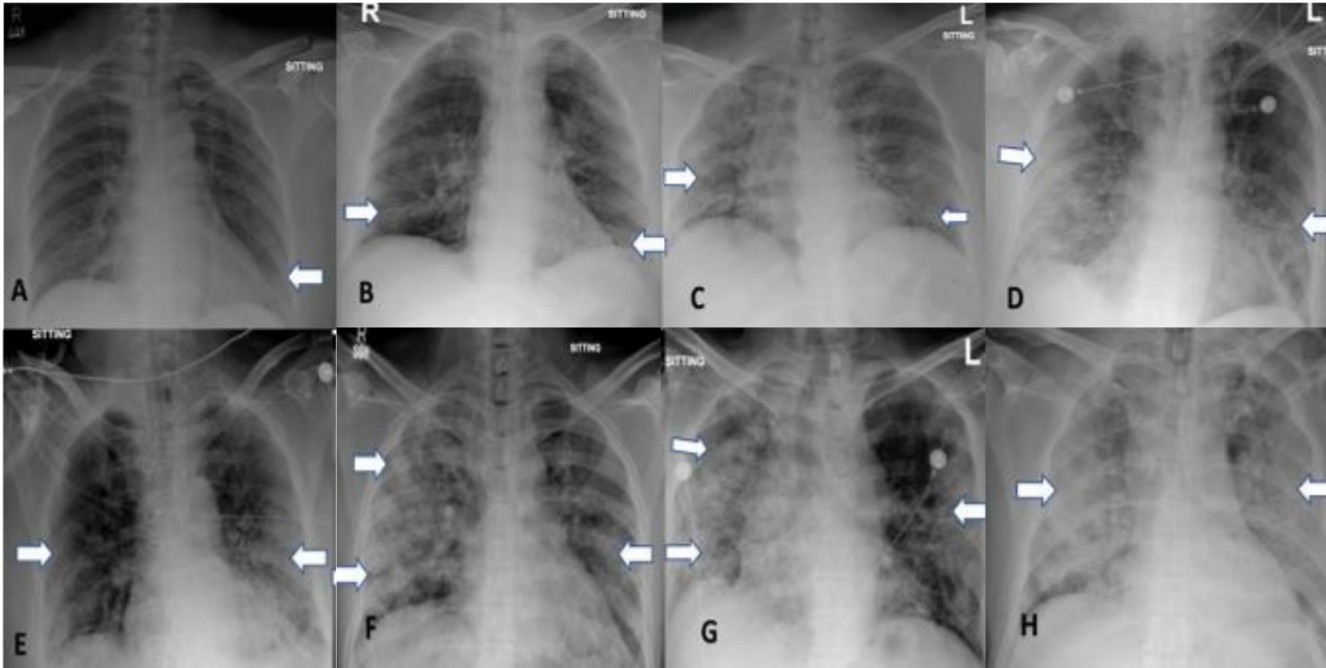

**Fig 1. CXR patterns.** A) left lower zone consolidation; B) bilateral lower zone interstitial infiltrates; F) scattered alveolar infiltrates in right lung; H) bilateral diffuse consolidation. CXR Severity Scoring (Right + Left): (A) 0+1 = 1; (B) 1+1 = 2; (C) 2+1 = 3; (D) 2+2 = 4; (E) 2+3 = 5; (F) 4+2 = 6; (G) 4+3 = 7; (H) 4 +4 = 8.

serial CXRs. Using the highest CXR-SS reduces heterogeneity, as it gives us a set standard of the maximum pulmonary involvement as seen on CXR to compare patients' outcomes against.

## Statistical analysis

All analyses were performed on IBM SPSS v. 21. Continuous data was presented as mean and standard deviation, and compared using independent sample t-tests. CXR Severity Scores were compared using Mann-Whitney U-Tests. Categorical data was presented as frequencies and percentages, and compared using Chi-squared tests. Spearman's correlations were used to investigate the correlations of continuous variables with CXR-SS. Univariate and multivariable logistic regression was performed with the dependent variables being initial CXR result (positive), initial CXR-SS (5–8), mortality, and ICU-Int. The multivariable models included age, gender, and variables with a p-value < 0.25 on univariate analysis. A p-value < 0.05 was considered significant for all analyses.

## Results

A total of 150 patients were included in this study, with the majority being male (76.7%). The mean age was 56.1 years, with ages ranging from 23–83 years. The commonest comorbids in our patients were hypertension (HTN: 46.7%) and diabetes mellitus (T2DM: 37.3%). A positive initial CXR result (CXR Severity Score > 0) was seen in 120 (80%) of patients. The mean initial CXR-SS was 3.32 ± 2.53. The majority of patients were admitted to the hospital (96%), and amongst these 30.6% were admitted to the ICU and intubated. While 72.2% of patients were healthy on discharge, 11.1% left without medical advice or were discharged on request. A mortality rate of 16.7% was observed in admitted patients. Patients with a positive initial CXR result were significantly older than those with a negative initial CXR result (58.5 ± 13.75 years

vs. 46.5 ± 17.13 years; p = 0.001), and also had a significantly greater highest CXR-SS (5.41 ± 2.23 vs. 2.39 ± 2.86; p < 0.001). Additionally, a significantly higher percentage of patients with positive initial CXR results were admitted in the ICU and intubated, as compared to those with negative initial CXR (34.5% vs. 14.3%; p = 0.037). The initial CXR-SS was also the highest CXR-SS in 56% of patients, while the highest CXR-SS of the remaining patients was seen on subsequent serial CXRs. In our study, 15 (10%) of patients were labeled with the code "do-not-resuscitate" (DNR), whilst the rest were labeled full-code. Amongst the patients labeled DNR, 10 (66.7%) expired and 3 (20%) were discharged on request/left against medical advice. The patients labeled DNR had a significantly greater mean age (63.40 ± 15.47 years vs. 55.32 ± 14.84 years; p = 0.049), initial CXR-SS (5.00 ± 2.56 vs. 3.19 ± 2.47; p = 0.008), and highest CXR-SS (6.87 ± 1.64 vs. 4.59 ± 2.64; p < 0.001) than those labeled full code. Details of patients' demographics and hospital course are shown in Table 1.

Males had a significantly greater highest CXR-SS than females (4.98 ± 2.66 vs. 3.83 ± 2.62; p = 0.023), as well as a higher mortality rate, though this was not statistically significant (20% vs. 5.9%; p = 0.054).

The majority of patients had an initial CXR-SS between 0–4 (69.3%), whereas 52% of patients had a highest CXR-SS between 5–8. On initial CXR, the vast majority of patients showed bilateral (92%) infiltrates, and these were peripherally located in 81.3% of patients. Infiltrates on initial CXR were most commonly located in the lower zone (72%) and showed features of consolidations (80%). The radiographic features of patients in our study are shown in Table 2.

**Table 1. Association of patients' characteristics and hospital course with initial CXR result.**

| Variable | Overall (N = 150) n (%)/Mean ± SD | Initial CXR Result | | P-Value |
|---|---|---|---|---|
| | | Positive (N = 120) n (%)/Mean ± SD | Negative (N = 30) n (%)/Mean ± SD | |
| Age (years) | 56.1 ± 15.21 | 58.5 ± 13.75 | 46.5 ± 17.13 | **0.001** |
| Gender | | | | |
| Male | 115 (76.7) | 94 (78.3) | 21 (70.0) | 0.334 |
| Female | 35 (23.3) | 26 (21.7) | 9 (30.0) | |
| Highest CXR-SS | 4.71 ± 2.68 | 5.41 ± 2.23 | 2.39 ± 2.86 | **< 0.001** |
| HTN | 70 (46.7) | 59 (49.2) | 11 (36.7) | 0.220 |
| T2DM | 56 (37.3) | 48 (40.0) | 8 (26.7) | 0.177 |
| CKD | 7 (4.7) | 7 (5.8) | 0 (0) | 0.346 |
| Malignancy | 1 (0.7) | 1 (0.8) | 0 (0) | > 0.999 |
| Initial RT-PCR | | | | |
| Positive | 141 (94.0) | 113 (94.2) | 28 (93.3) | > 0.999 |
| Negative | 9 (6.0) | 7 (5.8) | 2 (6.7) | |
| CT Done | | | | |
| Yes | 41 (27.3) | 35 (29.2) | 6 (20.0) | 0.314 |
| No | 109 (72.7) | 85 (70.8) | 24 (80.0) | |
| Admission Status | | | | |
| Inpatient | 144 (96.0) | 116 (96.7) | 28 (93.3) | 0.345 |
| Outpatient | 6 (4.0) | 4 (3.3) | 2 (6.7) | |
| ICU with Intubation | **N = 144** | **N = 116** | **N = 28** | **0.037** |
| Yes | 44 (30.6) | 40 (34.5) | 4 (14.3) | |
| No | 100 (69.4) | 76 (65.5) | 24 (85.7) | |
| Outcome | **N = 144** | **N = 116** | **N = 28** | |
| Recovered | 104 (72.2) | 81 (69.8) | 23 (82.1) | 0.252 |
| LAMA/DOR | 16 (11.1) | 13 (11.2) | 3 (10.7) | |
| Mortality | 24 (16.7) | 22 (19.0) | 2 (7.1) | |
| LOS (days) | 10.55 ± 7.83 | 10.71 ± 7.65 | 9.89 ± 8.89 | 0.625 |

Compared to those with an initial CXR-SS 0–4, patients with an initial CXR-SS 5–8 had a significantly greater highest CXR-SS (6.96 ± 1.17 vs. 3.72 ± 2.56; p < 0.001), a higher rate of ICU-Int (48.9% vs. 22.2%; p = 0.001), and a higher mortality rate (28.9% vs. 11.1%; p = 0.008). Compared to those with a highest CXR-SS 0–4, patients with a highest CXR-SS 5–8 were significantly more likely to be male (84.6% vs. 68.1%; p = 0.017) and have diabetes (46.2% vs. 27.8%; p = 0.020), and had a significantly higher rate of ICU-Int (44.2% vs. 14.9%; p < 0.001), mortality (24.4% vs. 6.9%; p = 0.004), and hospital LOS (12.26 ± 8.36 vs. 8.58 ± 6.84; p = 0.005). The ICU-Int and mortality rates according to CXR Severity Scores are shown in **Fig 2**.

## Comorbids, CXR severity score and clinical outcomes

Compared to patients without diabetes, a significantly greater percentage of patients with diabetes underwent ICU-Int (43.6% vs. 22.5%; p = 0.007) and suffered mortality (27.3% vs. 10.1%; p = 0.007). Moreover, patients with diabetes also had a significantly greater age (62.6 ± 11.42 years vs. 52.2 ± 15.90 years; p < 0.001), initial CXR-SS (3.96 ± 2.52 vs. 2.94 ± 2.47; p = 0.015), highest CXR-SS (5.70 ± 2.13 vs. 4.13 ± 2.81; p < 0.001), and hospital LOS (12.53 ± 8.81 days vs. 9.33 ± 7.03 days; p = 0.025), compared to patients without diabetes.

Patients with hypertension had a significantly greater age (62.5 ± 12.19 years vs. 50.46 ± 15.42 years; p < 0.001), highest CXR-SS (5.31 ± 2.31 vs. 4.19 ± 2.89; p = 0.010), and hospital LOS (12.06 ± 8.44 days vs. 9.16 ± 7.11; p = 0.012), compared to patients without hypertension.

## Spearman's correlation of CXR Severity Scores (CXR-SS)

Initial CXR-SS demonstrated a weak but significant positive correlation with age (r = 0.203; p = 0.013), while the highest CXR-SS also demonstrated a weak but significant positive correlation with age (r = 0.211; p = 0.010) in addition to a moderate positive correlation with hospital LOS (r = 0.324; p < 0.001).

**Table 2. Characteristics of radiographic findings.**

| Radiological Properties | Overall (N = 150) |
|---|---|
| | n (%) |
| **Initial CXR-SS Category** | |
| 0–4 | 104 (69.3) |
| 5–8 | 46 (30.7) |
| **Highest CXR-SS Category** | |
| 0–4 | 72 (48.0) |
| 5–8 | 78 (52.0) |
| **Initial CXR Laterality** | |
| Unilateral | 12 (8.0) |
| Bilateral | 138 (92.0) |
| **Initial CXR Centrality** | |
| Central | 28 (18.7) |
| Peripheral | 122 (81.3) |
| **Initial CXR Zonal Location** | |
| Lower | 108 (72.0) |
| Lower and Middle | 27 (18.0) |
| Diffuse | 15 (10.0) |
| **Initial CXR Type of Infiltrates** | |
| Alveolar/Interstitial | 30 (20.0) |
| Consolidations | (80.0) |

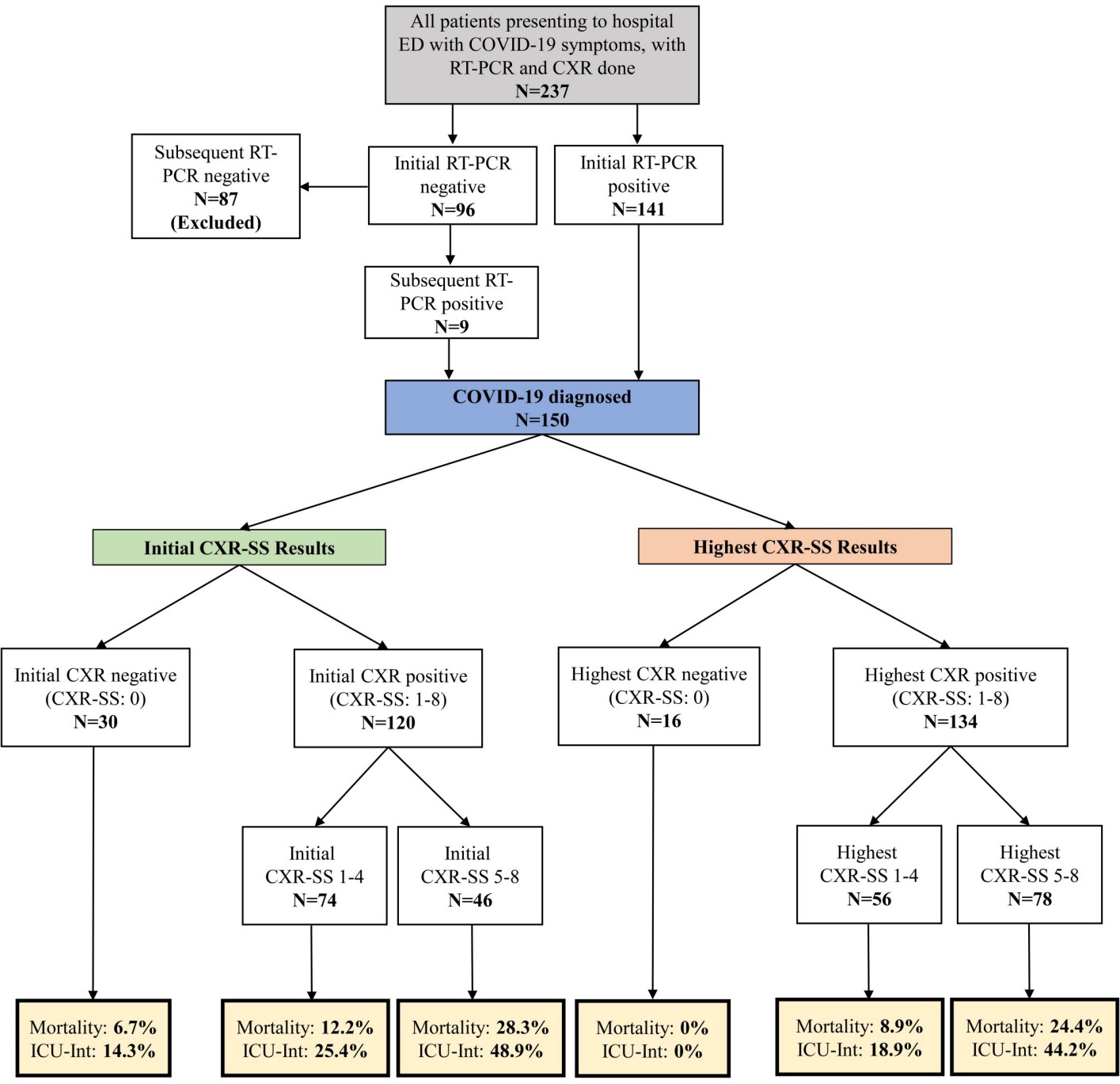

**Fig 2. Patient outcomes according to initial and highest Chest X-Ray Severity Score (CXR-SS).** ED: Emergency Department; ICU-Int: ICU Admission with Intubation; RT-PCR: reverse transcriptase polymerase chain reaction.

### Regression analysis for initial and highest CXR results (Table 3)

On univariate analysis, a positive initial CXR was associated with age (OR: 1.053 [95% CI: 1.022–1.085]) and ICU-Int (3.158 [1.025–9.733]). On multivariable analysis, only age was independently associated with a positive initial CXR (1.048 [1.014–1.083]).

**Table 3. Regression analyses for initial and highest CXR results.**

| Variable | 3A. Initial CXR Positive | | | |
|---|---|---|---|---|
| | cOR [95% CI] | P-Value | aOR [95% CI] | P-Value |
| Age (years) | 1.053 [1.022–1.085] | **0.001** | 1.048 [1.014–1.083] | **0.005** |
| Gender | | | | |
| Male | 1.724 [0.695–4.276] | 0.240 | 1.474 [0.555–3.919] | 0.437 |
| Female | Reference | - | Reference | - |
| T2DM | 1.703 [0.692–4.188] | 0.246 | 0.845 [0.303–2.354] | 0.747 |
| ICU with Intubation | 3.158 [1.025–9.733] | **0.045** | 2.085 [0.585–7.427] | 0.257 |
| Mortality | 3.043 [0.671–13.790] | 0.149 | 1.183 [0.214–6.527] | 0.847 |
| Variable | 3B. Initial CXR-SS 5–8 | | | |
| | cOR [95% CI] | P-Value | aOR [95% CI] | P-Value |
| Age (years) | 1.020 [0.995–1.045] | 0.120 | 1.006 [0.978–1.034] | 0.681 |
| Gender | | | | |
| Male | 2.036 [0.812–5.106] | 0.130 | 1.764 [0.673–4.626] | 0.249 |
| Female | Reference | - | Reference | - |
| T2DM | 1.673 [0.816–3.429] | 0.160 | 1.207 [0.538–2.703] | 0.648 |
| ICU with Intubation | 3.348 [1.578–7.105] | **0.002** | 2.532 [1.109–5.782] | **0.027** |
| Mortality | 3.250 [1.323–7.987] | **0.010** | 1.765 [0.638–4.881] | 0.274 |
| Variable | 3C. Highest CXR-SS 5–8 | | | |
| | cOR [95% CI] | P-Value | aOR [95% CI] | P-Value |
| Age (years) | 1.020 [0.998–1.043] | 0.078 | 1.004 [0.978–1.030] | 0.756 |
| Gender | | | | |
| Male | 2.648 [1.190–5.891] | **0.017** | 2.426 [1.027–5.731] | **0.043** |
| Female | Reference | - | Reference | - |
| T2DM | 1.958 [0.983–3.901] | 0.056 | 1.442 [0.659–3.155] | 0.360 |
| ICU with Intubation | 4.507 [2.008–10.117] | **< 0.001** | 3.386 [1.405–8.159] | **0.007** |
| Mortality | 4.062 [1.424–11.587] | **0.009** | 1.807 [0.560–5.827] | 0.322 |

On univariate analysis, initial CXR-SS 5–8 was associated with ICU-Int (3.348 [1.578–7.105]) and mortality (3.250 [1.323–7.987]). However, on multivariable analysis, only ICU-Int was independently associated with an initial CXR-SS 5–8 (2.532 [1.109–5.782]).

On univariate analysis highest CXR-SS 5–8 was associated with male gender (2.648 [1.190–5.891]), ICU-Int (4.507 [2.008–10.117]), and mortality (4.062 [1.424–11.587]). On multivariable analysis, however, CXR-SS 5–8 was independently associated with male gender (2.426 [1.027–5.731]) and ICU-Int (3.386 [1.405–8.159]).

## Regression analysis for mortality and ICU admission with intubation (Table 4)

On univariate logistic regression for mortality, age (1.063 [1.024–1.104]), initial CXR-SS (1.350 [1.119–1.628]), highest CXR-SS (1.406 [1.130–1.750]), T2DM (3.333 [1.343–8.276]) and ICU-Int (8.365 [3.143–22.265]) were associated with mortality. On multivariable logistic regression, age (1.060 [1.009–1.113]), initial CXR-SS (1.278 [1.010–1.617]) and ICU-Int (5.047 [1.731–14.710]) were found to be independent predictors of mortality.

On univariate logistic regression for ICU-Int, age (1.039 [1.012–1.067]), initial CXR-SS (1.328 [1.139–1.548]), highest CXR-SS (1.460 [1.224–1.742]), T2DM (2.671 [1.288–5.538]) and LOS (1.078 [1.029–1.129]) were associated with mortality. On multivariable logistic regression, initial CXR-SS (1.355 [1.136–1.616]) and highest CXR-SS (1.390 [1.143–1.690]) were found to

Table 4. Regression analyses for mortality and ICU admission with intubation.

| Variable | Mortality | | | |
|---|---|---|---|---|
| | cOR [95% CI] | P-Value | aOR [95% CI] | P-Value |
| Age (years) | 1.063 [1.024–1.104] | **0.001** | 1.060 [1.009–1.113] | **0.021** |
| Gender | | | | |
| Male | 4.000 [0.890–17.981] | 0.071 | 4.011 [0.762–21.108] | 0.101 |
| Female | Reference | - | Reference | - |
| Initial CXR-SS | 1.350 [1.119–1.628] | **0.002** | 1.278 [1.010–1.617] | **0.041** |
| Highest CXR-SS | 1.406 [1.130–1.750] | **0.002** | 1.250 [0.964–1.621] | 0.092 |
| T2DM | 3.333 [1.343–8.276] | **0.009** | 1.763 [0.484–6.417] | 0.390 |
| HTN | 2.037 [0.827–5.017] | 0.112 | 1.096 [0.279–4.313] | 0.895 |
| ICU with Intubation | 8.365 [3.143–22.265] | **< 0.001** | 5.047 [1.731–14.710] | **0.003** |
| Variable | ICU Admission with Intubation | | | |
| | cOR [95% CI] | P-Value | aOR [95% CI] | P-Value |
| Age (years) | 1.039 [1.012–1.067] | **0.005** | 1.031 [0.999–1.064] | 0.058 |
| Gender | | | | |
| Male | 1.581 [0.651–3.838] | 0.311 | 1.149 [0.425–3.107] | 0.785 |
| Female | Reference | - | Reference | - |
| Initial CXR-SS | 1.328 [1.139–1.548] | **< 0.001** | 1.355 [1.136–1.616] | **0.001** |
| Highest CXR-SS | 1.460 [1.224–1.742] | **< 0.001** | 1.390 [1.143–1.690] | **0.001** |
| T2DM | 2.671 [1.288–5.538] | **0.008** | 1.520 [0.661–3.495] | 0.324 |
| LOS (days) | 1.078 [1.029–1.129] | **0.002** | 1.082 [1.026–1.142] | **0.003** |

be an independent predictor of ICU-Int. A longer LOS was also independently associated with ICU-Int (1.082 [1.026–1.142]).

## Discussion

Early prognostication of disease remains a prevailing challenge in the ongoing COVID-19 pandemic, especially in developing countries where healthcare resources are limited. Previously published data from China and the developed world has highlighted the potential role of imaging in the early identification and prognostication of COVID-19 [11, 13–15]. In this study, we described the experience of CXR imaging in patients with COVID-19 at a tertiary care hospital in Pakistan, and explored the association of a CXR-SS proposed by Wong et al. [11] with clinical outcomes such as ICU admission with intubation (ICU-Int) and mortality. Though previous studies have assessed the value of various CXR scoring systems in the management of patients with COVID-19 [11, 13–15], to the best of our knowledge this is the first study done using a CXR-SS adapted from the RALE score in prognostication of clinical outcomes of COVID-19 patients.

The majority of patients in our study had bilateral, peripheral disease with predominantly lower lobe distribution, with consolidation being the prominent feature, on initial CXR. This pattern of findings has been demonstrated previously [6, 16]. Lymphadenopathy, pleural effusion and pneumothorax were infrequent findings in our study. On initial CXR, the majority (69.3%) of patients had a CXR-SS between 0–4. However, 52% of patients' highest CXR-SS ranged from 5–8.

On multivariable logistic regression, a positive initial CXR (CXR-SS > 0) was associated with greater age (OR: 1.060 [95% CI: 1.009–1.113]), while initial CXR-SS 5–8 was associated with ICU-Int (2.532 [1.109–5.782]). Moreover, initial CXR-SS was also found to be an independent predictor of both ICU-Int (1.355 [1.136–1.616]) and mortality (1.278 [1.010–1.617]).

Patients' highest CXR-SS was also identified as a predictor of ICU-Int (1.390 [1.143–1.690]), with a highest CXR-SS 5–8 also being associated with the male gender (2.426 [1.027–5.731]). Thus, the CXR-SS system used in our study appeared to show a strong relationship with clinical outcomes, with greater CXR-SS being associated with ICU-Int (initial and highest CXR-SS) and mortality (initial CXR-SS). Moreover, ICU-Int was independently associated with a longer length of hospital stay (1.082 [1.026–1.142]). Other independent predictors of mortality included patient age (1.060 [1.009–1.113]) and ICU-Int (5.047 [1.731–14.710]). A study by Borghesi et al. similarly found that their self-designed CXR severity score (named the Brixia score) was independently associated with mortality [13, 14]. Toussie et al. similarly reported that their CXR severity score predicted hospital admission and intubation [15]. Additionally, the independent association of CXR-SS 5–8 with male gender in our study is also in line with observations whereby males are found to be disproportionately affected by COVID-19 [17]. The findings of our study, when taken in consideration collectively, hold particular relevance for resource-constrained countries such as Pakistan, where a longer length of hospital stay translates into additional expenses for the patient. The CXR-SS is useful as it is easily reproducible, and enables radiologists to provide treating physicians with understandable quantitative information regarding the extent of pulmonary infection. However, further studies with larger patient samples are required to validate the CXR-SS as a tool for prognostication in COVID-19.

Limitations of this study include the retrospective nature of its design, and the relatively small sample size. However, observer bias was addressed by blinding the radiologists to the diagnosis and clinical condition of the patient. Furthermore, findings on initial CXR were variable due to a heterogeneity in duration of symptoms and infection upon presentation to the hospital's emergency department. Moreover, as portable CXRs were used, positioning and other exposure-related factors may confound CXR findings. Lastly, this study did not contain a control group of patients (COVID-19 negative) to compare CXR findings and CXR-SS with.

## Conclusion

The chest x-ray severity scoring (CXR-SS) system used in this study is a valuable method of disease prognostication in COVID-19, as initial and highest subsequent CXR-SS show strong association with ICU admission and mortality. The benefits of this CXR-SS lie in its reproducibility, ability to convey easily understandable objective information between radiologist and treating physician, and feasibility in resource-constrained settings. While the results of our study serve as an initial step towards the validation of the CXR-SS as a prognosticative tool, further studies with larger samples are warranted in this regard.

## Author Contributions

**Conceptualization:** Raima Kaleemi, Kiran Hilal, Ainan Arshad, Haq TU, Sundas Basharat, Zeeshan Ansar.

**Data curation:** Raima Kaleemi, Ainan Arshad, Avinash Nankani.

**Formal analysis:** Russell Seth Martins.

**Funding acquisition:** Kiran Hilal.

**Project administration:** Kiran Hilal, Ainan Arshad.

**Software:** Russell Seth Martins.

**Supervision:** Kiran Hilal, Ainan Arshad.

**Validation:** Avinash Nankani.

**Writing – original draft:** Raima Kaleemi, Ainan Arshad, Russell Seth Martins, Avinash Nankani.

**Writing – review & editing:** Raima Kaleemi, Kiran Hilal, Ainan Arshad, Russell Seth Martins, Avinash Nankani, Haq TU, Sundas Basharat, Zeeshan Ansar.

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
