## [Decision Letter · Decision Letter 0]

25 Nov 2020

PONE-D-20-24338

The Association of Chest Radiographic Findings and Severity Scoring with Clinical Outcomes in Patients with COVID-19 Presenting to the Emergency Department of a Tertiary Care Hospital in Pakistan

PLOS ONE

Dear Dr. Kiran Hilal,

Thank you for submitting your manuscript to PLOS ONE. After careful consideration, we feel that it has merit but does not fully meet PLOS ONE’s publication criteria as it currently stands. Therefore, we invite you to submit a revised version of the manuscript that addresses the points raised during the review process.

We look forward to receiving your revised manuscript.

Kind regards,

Francesco Di Gennaro

Academic Editor

PLOS ONE

Journal Requirements:

Additional Editor Comments (if provided):

Dear Authors follow revieres suggestion to improve your article

Reviewers' comments:

Reviewer's Responses to Questions

**Comments to the Author**

1. Is the manuscript technically sound, and do the data support the conclusions?

Reviewer #1: Yes

Reviewer #2: Yes

2. Has the statistical analysis been performed appropriately and rigorously? 

Reviewer #1: Yes

Reviewer #2: Yes

3. Have the authors made all data underlying the findings in their manuscript fully available?

Reviewer #1: Yes

Reviewer #2: Yes

4. Is the manuscript presented in an intelligible fashion and written in standard English?

Reviewer #1: No

Reviewer #2: Yes

5. Review Comments to the Author

Reviewer #1: Thank you very much for your submission.

This is a nice study, which could set a nice ground for future research on prognosis and outcome of COVID-19.

As a general comment, language and grammar needs revision. It does not deter from the possibility to understand the text, but there are multiple typos.

I would also review the format.

The tables could be reviewed and perhaps combined, the format could be made homogeneous.

The resolution and quality of images should be reviewed .

Reviewer #2: This is an high quality article and I appreciate a lot the paper.

Only some minor suggestions:

- discuss the possible role of ultrasound already in use in other infectious diseases ex tuberculosis and HIV (see and cite Bobbio F, Focused ultrasound to diagnose HIV-associated tuberculosis (FASH) in the extremely resource-limited setting of South Sudan: a cross-sectional study. BMJ Open. 2019 Apr 2;9(4):e027179. doi: 10.1136/bmjopen-2018-027179. PMID: 30944140; PMCID: PMC6500283.)

- Gigure 2 is not esay to read for the quality of image. Please add image with high quality

6. PLOS authors have the option to publish the peer review history of their article (what does this mean?). If published, this will include your full peer review and any attached files.

Reviewer #1: **Yes: **SALMAN S. ALBAKHEET

Reviewer #2: No

---

## [Author Response · Author response to Decision Letter 0]

14 Dec 2020

1. Is the manuscript technically sound, and do the data support the conclusions? The manuscript must describe a technically sound piece of scientific research with data that supports the conclusions. Experiments must have been conducted rigorously, with appropriate controls, replication, and sample sizes. The conclusions must be drawn appropriately based on the data presented.

Reviewer #1: Yes

Reviewer #2: Yes

Authors’ Response: Thank you.

2. Has the statistical analysis been performed appropriately and rigorously?

Reviewer #1: Yes

Reviewer #2: Yes

Authors’ Response: Thank you.

3. Have the authors made all data underlying the findings in their manuscript fully available? The PLOS Data policy requires authors to make all data underlying the findings described in their manuscript fully available without restriction, with rare exception (please refer to the Data Availability Statement in the manuscript PDF file). The data should be provided as part of the manuscript or its supporting information, or deposited to a public repository. For example, in addition to summary statistics, the data points behind means, medians and variance measures should be available. If there are restrictions on publicly sharing data—e.g. participant privacy or use of data from a third party—those must be specified.

Reviewer #1: Yes

Reviewer #2: Yes

Authors’ Response: Thank you.

4. Is the manuscript presented in an intelligible fashion and written in standard English? PLOS ONE does not copyedit accepted manuscripts, so the language in submitted articles must be clear, correct, and unambiguous. Any typographical or grammatical errors should be corrected at revision, so please note any specific errors here.

Reviewer #1: No

Reviewer #2: Yes

Authors’ Response: Thank you for your comment. We have proofread our manuscript thoroughly and ensured that all the minor typographical and grammatical errors are corrected (highlighted in the text).

5. Review Comments to the Author Please use the space provided to explain your answers to the questions above. You may also include additional comments for the author, including concerns about dual publication, research ethics, or publication ethics. (Please upload your review as an attachment if it exceeds 20,000 characters)

Reviewer #1: Thank you very much for your submission. This is a nice study, which could set a nice ground for future research on prognosis and outcome of COVID-19. As a general comment, language and grammar needs revision. It does not deter from the possibility to understand the text, but there are multiple typos. I would also review the format. The tables could be reviewed and perhaps combined; the format could be made homogeneous. The resolution and quality of images should be reviewed .

Authors’ Response: Thank you for your appreciation and your feedback. We have proofread the manuscript thoroughly and corrected grammatical errors (highlighted in manuscript). We have also reviewed the format of Table 2 and shortened it by removing 2 rows (listing frequencies and percentages of the individual initial and highest CXR-SS) which we feel were redundant. Moreover, we have ensured the resolution of the images is 1200 dpi and have used the PACE tool to make sure they are appropriate. We hope that it is adequate now.

Reviewer #2: This is an high quality article and I appreciate a lot the paper. Only some minor suggestions: Discuss the possible role of ultrasound already in use in other infectious diseases ex tuberculosis and HIV (see and cite Bobbio F, Focused ultrasound to diagnose HIV-associated tuberculosis (FASH) in the extremely resource-limited setting of South Sudan: a cross-sectional study. BMJ Open. 2019 Apr 2;9(4):e027179. DOI: 10.1136/bmjopen-2018-027179. PMID: 30944140; PMCID: PMC6500283.) Figure 2 is not easy to read for the quality of image. Please add image with high quality

Authors’ Response: Thank you for your appreciation and your feedback. We have added the suggested Reference in our Introduction, and discussed the role of ultrasound as well: “The use of ultrasound (US) has been explored previously in the diagnostics of other infectious diseases (2). Some studies have considered the emerging role of portable US imaging of the lungs in COVID-19 detection, describing ultrasound findings such as B-lines, consolidation and thickened pleural lines (3). However, US has its limitations, as it is highly operator-dependent and difficult to perform in patients with severe disease in an intensive care unit (ICU) set-up. Moreover, the specificity of ultrasound in patients with COVID-19 presenting with early disease has also proven to be significantly lower as compared to chest x-ray and computerized tomography scans (4)”. Additionally, we have ensured the resolution of the images is 1200 dpi and have used the PACE tool to make sure they are appropriate. We hope that it is adequate now.

1. Thank you for including your ethics statement on the online submission form: "For this retrospective study, ethical approval was obtained from the institutional review board of the Aga Khan University Hospital (Reference Number: 2020-4774-10611). We were granted a waiver of informed consent since it was retrospective study and all those patients were discharged from the hospital. Records were anonymised to the statistician and X-rays were anonymised to the radiographers as to the clinical condition of patient. ". 

To help ensure that the wording of your manuscript is suitable for publication, would you please also add this statement at the beginning of the Methods section of your manuscript file.

Authors’ Response: Thank you. We have added the following to the first paragraph of our Methods section: “For this retrospective study, ethical approval was obtained from the institutional review board of the Aga Khan University Hospital (Reference Number: 2020-4774-10611). We were granted a waiver of informed consent as this was a retrospective study and all patients were discharged from the hospital. No personal identifiers were included in data collection, and records were anonymized to the statistician.”. We hope it is adequate.

2. Please ensure that you refer to Figure 1 in your text as, if accepted, production will need this reference to link the reader to the figure.

Authors’ Response: Thank you. We have added the following to our Methods section: “An example of the application of this scoring system is shown in Figure 1.”.

---

## [Decision Letter · Decision Letter 1]

18 Dec 2020

The Association of Chest Radiographic Findings and Severity Scoring with Clinical Outcomes in Patients with COVID-19 Presenting to the Emergency Department of a Tertiary Care Hospital in Pakistan

PONE-D-20-24338R1

Dear Dr. Hilal,

We’re pleased to inform you that your manuscript has been judged scientifically suitable for publication and will be formally accepted for publication once it meets all outstanding technical requirements.

Kind regards,

Francesco Di Gennaro

Academic Editor

PLOS ONE

Additional Editor Comments (optional):

dear authors congratulations

Reviewers' comments:

Reviewer's Responses to Questions

**Comments to the Author**

1. If the authors have adequately addressed your comments raised in a previous round of review and you feel that this manuscript is now acceptable for publication, you may indicate that here to bypass the “Comments to the Author” section, enter your conflict of interest statement in the “Confidential to Editor” section, and submit your "Accept" recommendation.

Reviewer #2: All comments have been addressed

2. Is the manuscript technically sound, and do the data support the conclusions?

Reviewer #2: Yes

3. Has the statistical analysis been performed appropriately and rigorously? 

Reviewer #2: Yes

4. Have the authors made all data underlying the findings in their manuscript fully available?

Reviewer #2: Yes

5. Is the manuscript presented in an intelligible fashion and written in standard English?

Reviewer #2: Yes

6. Review Comments to the Author

Reviewer #2: Authors wrote an interesting article on a topic issue form interesting setting. I apprecciate this new version and in my opinion is suistable to publish in this present form

7. PLOS authors have the option to publish the peer review history of their article (what does this mean?). If published, this will include your full peer review and any attached files.

Reviewer #2: No

---

## [Editor Report · Acceptance letter]

26 Dec 2020

PONE-D-20-24338R1 

The Association of Chest Radiographic Findings and Severity Scoring with Clinical Outcomes in Patients with COVID-19 Presenting to the Emergency Department of a Tertiary Care Hospital in Pakistan 

Dear Dr. Hilal:

I'm pleased to inform you that your manuscript has been deemed suitable for publication in PLOS ONE. Congratulations! Your manuscript is now with our production department. 

Kind regards, 

on behalf of

Dr. Francesco Di Gennaro 

Academic Editor

PLOS ONE